# Exploration of Driver Posture Monitoring Using Pressure Sensors with Lower Resolution

**DOI:** 10.3390/s21103346

**Published:** 2021-05-12

**Authors:** Mingming Zhao, Georges Beurier, Hongyan Wang, Xuguang Wang

**Affiliations:** 1School of Automotive Studies, Tongji University, Shanghai 201804, China; 1710085@tongji.edu.cn (M.Z.); why-sos@vip.sina.com (H.W.); 2Univ Lyon, Univ Gustave Eiffel, Université Claude Bernard Lyon 1, LBMC UMR_T9406, F69622 Lyon, France; georges.beurier@univ-eiffel.fr

**Keywords:** driver posture monitoring, pressure measurement, sensor layout, machine learning

## Abstract

Pressure sensors are good candidates for measuring driver postural information, which is indicative for identifying driver’s intention and seating posture. However, monitoring systems based on pressure sensors must overcome the price barriers in order to be practically feasible. This study, therefore, was dedicated to explore the possibility of using pressure sensors with lower resolution for driver posture monitoring. We proposed pressure features including center of pressure, contact area proportion, and pressure ratios to recognize five typical trunk postures, two typical left foot postures, and three typical right foot postures. The features from lower-resolution mapping were compared with those from high-resolution Xsensor pressure mats on the backrest and seat pan. We applied five different supervised machine-learning techniques to recognize the postures of each body part and used leave-one-out cross-validation to evaluate their performance. A uniform sampling method was used to reduce number of pressure sensors, and five new layouts were tested by using the best classifier. Results showed that the random forest classifier outperformed the other classifiers with an average classification accuracy of 86% using the original pressure mats and 85% when only 8% of the pressure sensors were available. This study demonstrates the feasibility of using fewer pressure sensors for driver posture monitoring and suggests research directions for better sensor designs.

## 1. Introduction

Human driver errors in terms of attention, recognition, decision, and performance have been reportedly regarded as the main causes of traffic accidents by various sources [1,2,3]. Although driving automation is beneficial for accommodating human errors, a driver’s unavailability to take over may pose new challenges to road traffic safety [4,5,6]. The postural information associated with the human driver (e.g., head orientation, positions of hands and feet relative to vehicle controls) should be continuously monitored for a safe and smooth transition [7,8]. From the perspective of vehicle passive safety, drivers in autonomous vehicles may adopt postures quite different from the standard driving posture [9]. In cases of unavoidable collisions, traditional restraint systems calibrated for proper positioning of the driver may not provide efficient protection [10]. In order to reduce the potential injuries, the automobile community is striving for intelligent restraint systems such as smart airbags, for which the collision responses can be modulated according to driver’s instantaneous seating position [9,11].

The purpose of driver posture monitoring is to provide fundamental information for evaluating the driver’s attention, operational intention, seating posture and position, and so on. This information can be incorporated into the development of driving automation systems and intelligent restraint systems for safer driving. 

Many existing postural monitoring methods are vision-based for capturing head or hand movement, such as RGB cameras and depth cameras [12]. Another noninvasive alternative or complementary approach is to detect the driver’s posture using pressure sensors embedded into driver seat. This technique has been explored by a few researchers in the field of automotive safety in an attempt to detect the driver’s state, seating position, or foot behavior [13,14,15]. However, the methods proposed by these pilot studies were not well validated on new drivers [16]. In the context of in-vehicle monitoring, the body pressure distribution (BPD) pattern is not only dependent on posture, but also affected by the driver’s anthropometric dimensions, as well as by individual seating preferences in terms of backrest angles, seat height, seat position, seat pan inclination, etc. To leverage the pressure sensors for in-vehicle monitoring, it is necessary to gain a thorough understanding of the relationship between the BPD and driver’s postures.

In our previous work [16], we used two Xsenor pressure mats with a resolution of 42 by 44, one on the seat pan and one on the backrest, to measure the BPD for 42 in-vehicle activities performed by 23 participants. Nine different postures characterized by typical trunk (normal position and five abnormal positions) and foot positions were defined for classification. Three different classifiers, including a deep learning classifier and two random forest (RF) classifiers were trained on raw BPD, as well as on absolute and relative pressure features with respect to a standard reference posture. The pressure features were extracted from segmented sensing areas, and their importance was evaluated by estimating the RF out-of-bag (OOB) errors [17]. The performance of the classifiers was evaluated by leave-one-out (LOO) cross-validation. Results showed that the RF classifier trained on relative pressure features achieved an average accuracy of 80.5% and outperformed the other two classifiers. Apart from further refinement of feature selection, an issue of using existing pressure sensor products is the price barrier for their implementation in commercialized vehicles. For example, the two Xsensor pressure mats employed in our work cost more than 10,000 EUR, which are too expensive to be a practical solution.

In recent years, low-cost BPD measurement systems have given rise to various applications for identifying the user’s in-chair sitting postures in the area of personal healthcare, human–computer interaction, and seat ergonomic design. For example, Bibbo et al. [18] instrumented a chair with four textile pressure sensors on the seat pan and four on the backrest to investigate if the postural variation is influenced by cognitive engagement. By referring to the initial posture and analyzing the pressure data over time, posture transitions between eight different postures could be successively detected. To detect the user’s posture in a wheelchair to reveal their sitting habit, Ma et al. [19] developed a posture recognition system with five pressure sensors on the backrest and seven on the seat pan. Similarly, five supervised classification techniques were compared, and the best performance was determined using the decision tree method. By using a backward selection method, the best sensor deployment was identified to have five sensors in total, achieving an accuracy of 99.47% for classifying five different postures by 10-fold cross-validation. Unlike previous studies [18,19], Roh et al. [20] implemented only four force sensors on the seat pan and used the pressure ratios (PRs) in anterior–posterior and medial–lateral directions to train various machine-learning algorithms for the automatic classification of six different sitting postures. The best performance was obtained for the support vector machine (SVM) classifier with an average classification accuracy of 97.2% for nine subjects. Bourahmoune and Amagasa [21] proposed a so-called LifeChair smart cushion with nine pressure sensor units only on the backrest for recognizing poor sitting postures. The sensors were first calibrated by a standard upright posture and the corresponding values stored as a reference frame. According to the pressure deviations with respect to the reference data, the RF classifier achieved an accuracy of 98.93% in detecting over 13 different sitting postures and 97.99% in detecting six common chair bound stretches from 10 subjects. 

To more accurately monitor in-chair activities, sophisticated systems based on the information from other sensors in addition to the BPD measurement have been developed. For example, Zemp et al. [22] tested five different machine-learning methods on the signal from 16 force sensors (10 on seat pan, four on backrest, and one on each armrest) and the backrest angle from a Motion-Module to distinguish seven different sitting positions. The best performance was achieved by the random forest (RF) classifier, which produced a mean classification accuracy of 90.9% by LOO cross-validation from 41 subjects. In one study performed by Ma et al. [23], the pressure sensors were accompanied by an inertial measurement unit (IMU) placed on the seat in order to capture the small movements of the body. The system utilized a sliding window to extract features from the sensors, and accuracies of 89% and 98% were achieved for activity recognition and activity level recognition, respectively. In another study [24], a system based on body-worn inertial sensors combined with six force sensors on the seat was designed by Gravina and Li to recognize four basic emotion-relevant activities performed by eight volunteers. By fusing the sensor- and feature-level features, an average classification accuracy of 91.8% was achieved. More recently, Jeong and Park [25] proposed a smart chair system consisting of six pressure sensors on the seat pan and six infrared reflective distance sensors on the backrest. Using a weighted k-nearest neighbor (k-NN) algorithm, 11 different sitting postures could be recognized with an average accuracy of 92% across 36 individuals. 

The results of previous studies [18,19,20,21,22,23,24,25] suggested that low-cost pressure sensors can possibly serve as the main or complementary part of a monitoring system to recognize different in-chair sitting postures with considerable accuracy. Although promising, it remains unclear if these methods are also valid for the monitoring of in-vehicle driver postures. Apparently, driver’s postural variation is certainly not as large as that of office chair users due to more restrained in-vehicle space and postural requirement for operating driving control commands. Monitoring small postural change (e.g., trunk rotation) is more challenging. More importantly, in-vehicle posture monitoring requires real-time performance, which is not necessarily important in the office environment. In addition, the sensor numbers and locations varied from study to study [18,19,20,21,22,23,24,25]. It seems that the appropriate number of sensors and their position are dependent on the types of chairs and the postures of interest.

Following our previous work [16], this study aimed to find a cost-effective solution for driver posture monitoring using only pressure sensors. Our main contributions are twofold. On one hand, more pressure feature candidates are extracted from the segmented BPD for capturing the movement of the driver’s local body parts. On the other hand, we propose a sampling method to design new pressure sensors with a lower resolution. We show that the number of sensors can be significantly reduced while keeping considerable posture recognition accuracy.

## 2. Materials and Methods

### 2.1. Data Collection

The experimental data from the previous work [16] were used for this study. Figure 1 shows an overview of our driver posture dataset. Here, we briefly review the data collection process.

Twenty-three differently sized males and females participated in our experiment. They were asked to perform 42 in-vehicle tasks, including primary driving tasks (e.g., braking, switching gear), secondary driving tasks (e.g., answering phone call, drinking coffee), non-driving-related tasks (e.g., reading books, sleeping), and general driver body actions (e.g., trunk rotation, arm flexion), on an experimental driving mockup. The standard posture (upright looking forward, right foot on the accelerator pedal, and left foot on the floor) was taken as both the initial and the final postures of each task.

The postural measurement (Figure 1) consisted of image flows from depth and RGB-D cameras (25 fps), motion capture data (mocap) by the VICON system (50 fps), and BPD data (25 fps) from two Xsensor pressure mats on the seat pan and backrest. The data acquisition process was electronically synchronized. The mocap data were used to reconstruct the joint angles and joint positions and served as the ground truth for data analysis.

### 2.2. Data Analysis

#### 2.2.1. Definition of Posture Classes

The reconstructed driver postures (Figure 2) were classified them into nine classes in our previous work [16] by concurrently considering trunk and foot positions. In the present work, we define postural classes by body part, making it possible to select more relevant pressure features for recognizing the posture of the trunk, as well as the left and right feet. The posture class definition scheme is illustrated in Figure 3. The trunk posture was characterized by three trunk angles (rotation, inclination, and lateral tilt) with respect to the standard trunk position (TP0) at the beginning of each task. By analyzing the trunk angles, four additional trunk posture classes (TP1 to TP4) were defined with noticeable deviations from TP0 in terms of rotation, inclination, or lateral tilt. For the right foot, three classes (RFP0—right foot on accelerator pedal, RFP1—right foot on brake pedal, and RFP2—right foot on floor) were defined, while two classes (LFP0—left foot on floor and LFP1—left foot on clutch) were defined for the left foot.

To reduce data redundancy and overfitting risk of the classifiers, an intra- and inter-motion filter was applied to remove similar postures of the same participant. Two trunk postures were regarded as different if one of the three trunk angles was higher than 3°. Two right-foot positions were regarded as different if the Euclidean distance between the foot centers was bigger than 2 cm. The same rule applied to the left foot. The filtering process was independently performed for each body part. Finally, 3999 trunk postures, 8024 right-foot postures, and 5216 left-foot postures were extracted from the experimental dataset.

#### 2.2.2. BPD Feature Extraction and Evaluation

According to the measured contact areas, 42 by 44 effective cells (1.27 × 1.27 cm^2^ for each cell) for both the seat pan and the backrest mats were determined. As in our previous work [16], the whole sensing area was divided into 12 (B1–B12) and eight (S1–S8) subareas for the backrest and seat pan, respectively (Figure 4) and 44 pressure parameters were defined. In the present work, we extended this feature collection to 200 candidates (Appendix A) by considering not only the global areas (B and S) and individual subareas, but also the regional areas composed of multiple adjacent subareas. 

As relative features with respect to the standard driving postures showed better prediction than absolute features from our previous study [16], only relative pressure parameters were used to evaluate the importance of pressure features using the OOB error estimation method proposed in [17] and to train the respective classifier for posture recognition.

#### 2.2.3. Posture Classification and Evaluation

In addition to the RF classifier [26], four other supervised machine-learning techniques, namely, support vector machine (SVM) [27], multilayer perceptron (MLP) [28], k-nearest neighbors (k-NN) [29], and naïve Bayes (NB) [28] were compared to find the best one for driver posture classification. The classifiers were implemented in Maltab 2020a, and their configurations are listed in Table 1.

In order to objectively evaluate the generalization capability of a classifier, LOO cross-validation was performed, where the data of all participants except one were used for training the classifier and the excluded one was used for validation. Iterating this procedure for each participant resulted in 23-fold cross-validation.

To quantify and evaluate the classification accuracy, the *F*1 *score* (a harmonic mean of *precision* and *recall*) was calculated from the confusion matrix. The *precision (PREC)* is referred to as the proportion of data samples that the classifier predicts true actually are true (Equation (1)), while the *recall (REC)* is referred to as the ability to predict the true results given the data samples of a specific class (Equation (2)). As opposed to the *accuracy* which is simply calculated as the number of all correct predictions divided by the total number of data samples, the *F*1 *Score* (Equation (3)) takes both false positives and false negatives into account and is, therefore, usually more useful, especially when the classes are unevenly distributed.
(1)$$PREC=\frac{TP}{TP+FP},$$
(2)$$REC=\frac{TP}{TP+FN},$$
(3)$$F1\;Score=\frac{2\cdot PREC\cdot REC}{PREC+REC},$$
where *TP* denotes the true positives, *FP* denotes the false positives, and *FN* denotes the false negatives.

#### 2.2.4. Pressure Sensor Layout Designs

To test if the number of pressure sensors could be reduced to overcome the price barrier, posture classification was performed on the basis of five newly designed pressure mats (D1–D5, Figure 5) by lowering the number of sensing elements compared to the original mats. These new layouts were designed using a uniform sampling method for both seat pan and backrest pressure mats (Figure 5). D1 was equipped with 13 sensors (one at each corner, one at the middle of each side, one at the center of each quadrant, and one at the center of the whole mat). Based on D1, we added extra points to the center of the hypotenuse of each triangle formed by three adjacent key points to generate D2 (25 sensors). In a similar manner, the new designs D3 (41 sensors), D4 (81 sensors), and D5 (145 sensors) were obtained by adding extra key points to the previous one. We preferred a uniform sampling method in an attempt to keep the features identified from initial high-resolution mats.

The new layouts were then overlaid onto the original full mats to extract pressure values. Note that some of the key points on the new layouts may not exactly lie at the center of one cell on the original mat. In these cases, the pressure value was read from the closest cell relative to the key point. After extracting the pressure values, an interpolation method based on inverse distance weighting [30] was performed to reconstruct the BPD. Finally, the important features identified for the original full mats were recalculated from the reconstructed BPD to train and test the classifiers. The posture classification accuracy based on the original pressure mats was regarded as benchmark. To determine if there was a significant difference between the results from the newly designed layouts and the original full mats, a paired *t*-test was performed.

## 3. Results

### 3.1. Evaluation of Importance of Pressure Parameters

Using the average F1 score across all posture classes from each body part as a metric, the best combination of pressure features for each RF classifier was determined. Figure 6 shows that the classifier RF-trunk achieved the highest prediction accuracy (91%) when only 27 important features were used, and the accuracy was notably compromised when more than 40 features were involved. The prediction accuracy of RF-leftFoot plateaued (93%) at around 24 important features and remained stable thereafter. For classifier RF-rightFoot, the potential best accuracy (around 74%) was first approached when 22 important features were selected. Note that the use of more features did not necessarily improve posture recognition scores. The selected important features are given in Appendix A.

### 3.2. LOO Cross-Validation of the Posture Classifiers

With the best combinations of pressure features extracted from the original pressure mats, the F1 scores from the 23-fold cross-validation of different classifiers are summarized in Table 2. The RF classifier outperformed the other four classifiers regardless of which body part or posture class was used. Using the three RF classifiers, F1 scores ranged from 85% to 98%, from 90% to 97%, and from 61% to 92% for the trunk, left-foot, and right-foot posture classes, respectively. An average F1 score of 86% across the 10 posture classes was obtained. The confusion matrices of the three RF classifiers are given in Figure 7. Among the three body parts, the main confusion was determined for the classification of right-foot postures. As shown in Figure 7b, RFP0 (right foot on accelerator pedal) was occasionally misclassified as RFP1 (right foot on brake pedal) or RFP2 (right foot on floor). 

### 3.3. Evaluation of the Proposed Pressure Sensor Layouts of Lower Resolution

Table 3 shows the F1 scores of posture classification with the RF classifiers for the five proposed pressure sensor layouts with different resolutions. Significant differences were found between the sensor layouts (D1, D2, D3, D4) and the original pressure mats. For D5 (145 × 2 sensors) with the highest resolution among the new layouts, only the recognition of TP4, LFP1, RFP1, and RFP2 showed significant difference, and the F1 scores were close to those achieved using the original pressure mats. Using the layout D5, an overall F1 Score of 85%, slightly lower than 86% for the original pressure mats, was obtained. Meanwhile, the sensor number was reduced by 92% ((44 × 42 × 2 – 145 × 2)/44 × 42 × 2), suggesting the possibility of using lower-resolution pressure mats for driver posture monitoring.

## 4. Discussion and Conclusions

The aim of this study was to provide insights into the development of a reliable yet pragmatic driver posture monitoring system using only pressure sensors. Results showed that the number of sensing elements could be reduced by 92% from the original pressure mats by using RF classifiers at the expense of a slight loss in posture recognition accuracy. This result is expected because the pressure features were extracted from regional sensing areas instead of individual sensing elements. Pressure features are not too sensitive to the mapping resolution unless too few sensors are available.

Compared to our previous work [16], we proposed a more systematic method for extracting and evaluating important pressure features. As opposed to the use of the 15 important features by the single RF classifier in the previous work, this work selected 40, 24, and 22 important features to classify trunk postures, left-foot postures, and right-foot postures, respectively. Thanks to the enrichment of important pressure features and the individualized posture recognition for each body part, we obtained an average F1 score of 86% across 10 posture classes by using the original pressure mats and RF classifiers. Even with the new pressure design D5 with only 145 × 2 sensing elements, an average F1 score of 85% was achieved, higher than the average F1 score of 80.5% across nine posture classes achieved in the previous work. Overall, the different trunk postures could be recognized with considerable accuracy (91% vs. 90% by original mats and new design D5). The main errors are still related to the recognition of two right-foot postures, RFP1 and RFP2, for which relatively low F1 scores of 70% and 61%, respectively, were obtained even when using the full original pressure mats. This is because foot movements (e.g., leg extension and lateral movement) are much more subtle than trunk movements. Therefore, the pressure variations caused by the former are much smaller than the latter. Another factor leading to low recognition performance of foot posture recognition is related to the behavioral difference across individuals, which was addressed in our previous work [16]. To overcome this issue, more data need to be collected to refine foot position classes, while the use of additional pressure sensors on the floor could also be of help.

When it comes to body pressure monitoring system, one attendant issue concerns the “BMI divergence”, which has been repeatedly mentioned in previous studies [19,21]. To investigate the effects of participant BMI on monitoring performance using our proposed system, we built a regression model between classification accuracy and BMI. However, no clear relationship was found, suggesting that the proposed system has robust performance across drivers of different body sizes. This could be, in large part, attributed to the relative features used by this study, which could reduce the effects of body size, as well as seat configuration, on the BPD.

This study also compared the performance of different machine learning methods (SVM, MLP, k-NN, and NB) for driver posture classification. The results revealed that the RF classifier outperformed the others and, therefore, is a suitable choice to deal with our pressure features.

Finding the optimum design of pressure sensors for driver posture monitoring was not the main focus of this study. This is a complex task requiring not only domain expertise but also trial and error. To test if high posture recognition accuracy could be maintained when reducing the resolution of the pressure sensors, we used a uniform sampling method. The motivation behind this choice is mainly related to our pressure feature engineering process, which was performed on the whole BPD from seat pan and backrest. Such a sampling method enabled reconstructing the BPD by interpolation. The use of interpolation, on the other hand, enables easily implementing the proposed sensor designs to new driver seat, because the exact match in terms of sensor location is not necessarily required.

According to the results revealed by the current study, several possible facets could be identified for exploring better pressure sensor layouts in terms of posture recognition performance and cost. First of all, the desirable layout is dependent on the posture classes to be recognized. As shown in Table 3, the accuracy of postural classification was affected differently by the reduction in sensor numbers. In particular, the worst case was found for TP4, which was quite similar to TP1 with a slight difference in the range of lateral tilt. The similarity between these two classes could explain the poor results obtained for TP4. Clearly, merging TP1 and TP4 as one class would allow more space by reducing the sensor number at the sacrifice of more distinctive posture descriptions. Furthermore, if one is only interested in distinguishing abnormal posture (trunk out of position, both feet on floor) from normal posture (standard trunk posture, right foot on the pedals, left foot on the floor or the clutch), the potential for reducing the sensor number will also become less constrained because fewer classes will facilitate the classification task.

Secondly, the pressure feature selection process could be further improved for better posture classification given a new layout with fewer sensors. In the current work, the same best pressure features were adopted for evaluating all sensor layouts. With a reduction in the number of sensor elements, the order of the importance of these feature candidates may be affected. New best pressure features may, thus, need to be reselected from the feature candidates calculated from the new pressure sensor design.

Thirdly, the sensor design on the backrest and seat pan was treated identically in this work. As indicated by the important features in Appendix A, the BPDs on the backrest and seat pan have different impacts on the posture recognition of each body part. Therefore, further effort is needed to individually investigate the suitable sensor number for the backrest and seat pan and to find the best combination of sensor designs.

In summary, this study demonstrated the feasibility of using only a BPD measurement system with fewer sensor elements for driver posture monitoring and suggests research directions for better pressure sensor designs to further reduce the economic cost. The proposed method also provides insights into the development of a cost-effective pressure sensors based system for monitoring in-chair activities in the office environment.

However, this study was subject to some limitations. First, the experiment was performed in a laboratory. In real driving conditions, the unevenness of the road, vehicle acceleration, and vibrations may challenge the robustness of our system. To this end, an extra inertial measurement unit (IMU) may be needed to compensate for vehicle acceleration and noise [15]. Second, the feasibility and effectiveness of the proposed method on different driver seats in different vehicles is somewhat unknown. Therefore, further validation is needed. In addition, using the proposed BPD measurement system alone cannot provide a holistic description of the driver’s posture. This can be supported by recent studies [22,23,24,25], where the pressure sensors served as a complementary part of the monitoring systems. Last but not least, the driver’s head posture and hand positions were beyond the detection range of pressure sensors on the seat. Recently, the advancement of computer vision technologies has allowed capturing driver body postures in detail [31]. However, researchers are still struggling with the posture recognition errors caused by body occlusions in the field of view. One possible solution is to build conditional posture recognition models based on some latent variables such as the trunk rotation relative to the camera [32]. In this circumstance, the trunk postures predicted by pressure sensors could be used as a priori knowledge to help the camera to achieve better performance. Inspired by previous studies [22,23,24,25,32], future work will be extended to use cameras in conjunction with pressure sensors to monitor driver posture and, of course, to test our method on a real road.

## Figures and Tables

**Figure 1 sensors-21-03346-f001:**
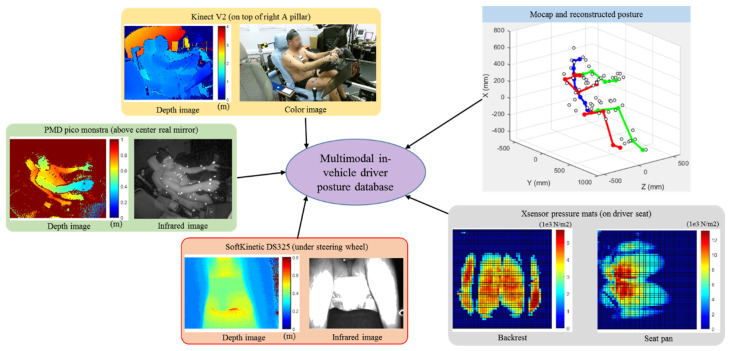
Driver posture dataset.

**Figure 2 sensors-21-03346-f002:**
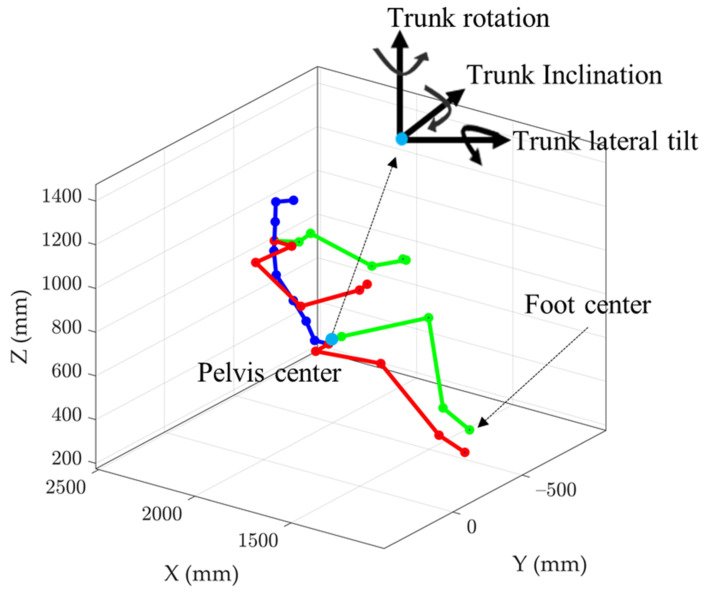
Driver posture after reconstruction using mocap data. A coordinate system located at the driver’s hip center was used to describe different trunk positions.

**Figure 3 sensors-21-03346-f003:**
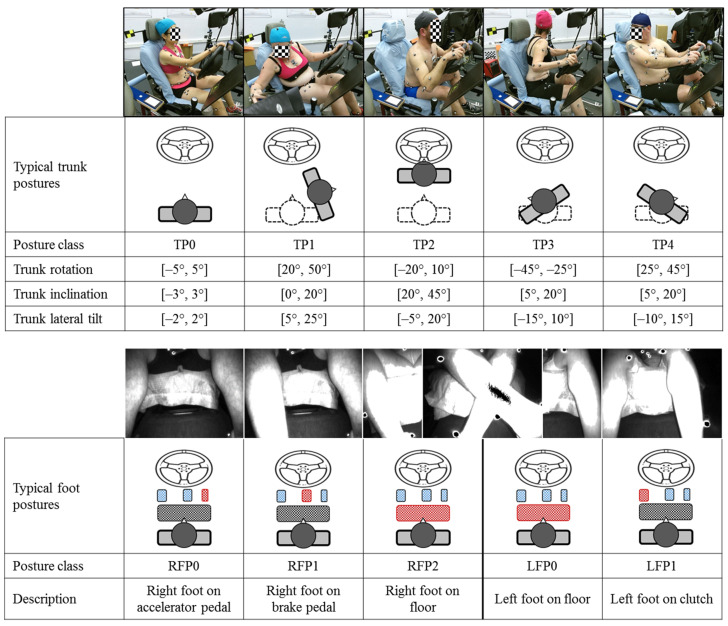
Definition of posture classes for trunk, right foot, and left foot.

**Figure 4 sensors-21-03346-f004:**
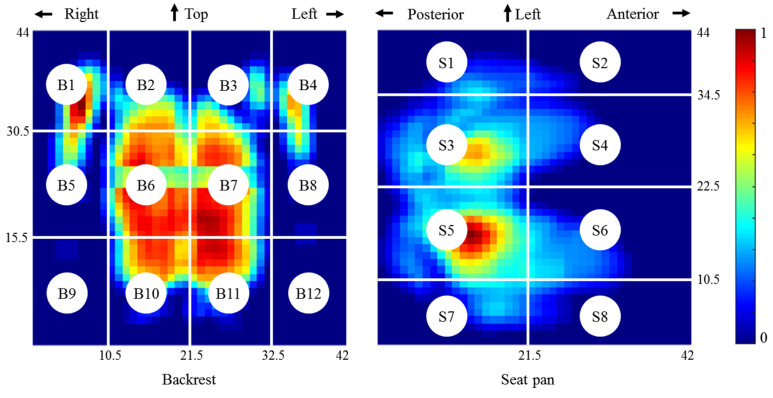
Pressure mat segmentation. The pressure distributions on the backrest (left) and seat pan (right) were normalized by the respective peak pressure.

**Figure 5 sensors-21-03346-f005:**
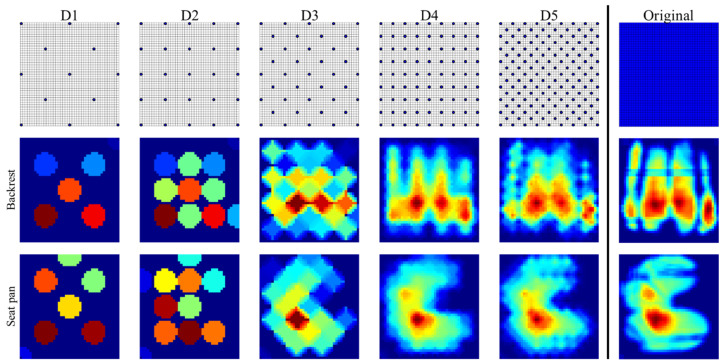
New pressure sensor layouts and interpolated BPD for backrest and seat pan. The first row shows different pressure sensor layouts, while the interpolated pressure distributions for backrest and seat pan are shown in the second row and the third row, respectively.

**Figure 6 sensors-21-03346-f006:**
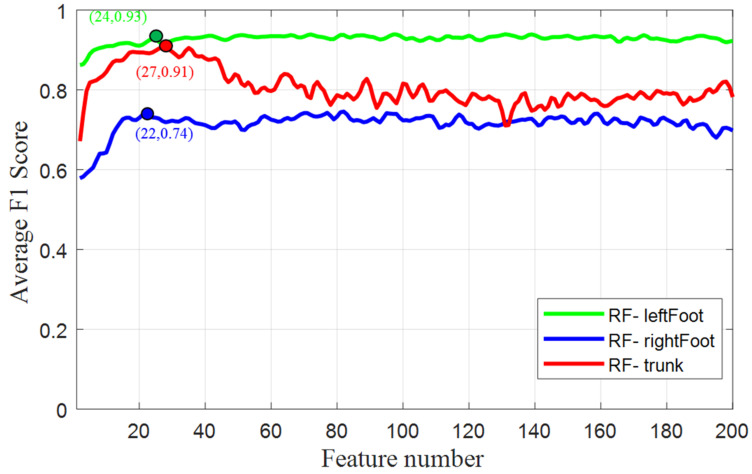
Average F1 score vs. feature number.

**Figure 7 sensors-21-03346-f007:**
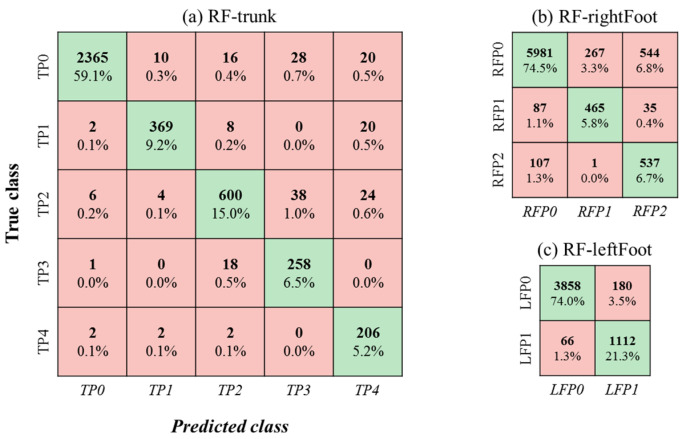
Confusion matrices of three RF classifiers. The green downward diagonal shows the number and proportion of correct detection cases for each class.

**Table 1 sensors-21-03346-t001:** Summary of configurations of classifiers.

Classifier	Parameters	Value
RF	Number of variables to sample	10
	Maximum number of splits	200
	Predictor selection criterion	“interaction curvature”
	Other parameters	Default
SVM	Model	error-correcting output code multiclass
	Kernel function	“rbf”
	Kernel scale	“auto”
	Standardize	true
	Other parameters	Default
MLP	Number of hidden layers	512
	Size of mini batch	300
	Optimizer	“adam”
	Maximum number of epochs	40
	Other parameters	Default
k-NN	Number of neighbors	3
	Other parameters	Default
NB	Data distributions	Multivariate multinomial distribution
	Other parameters	Default

**Table 2 sensors-21-03346-t002:** Classification results with the best combination of features based on original pressure mats.

Class	RF	NB	SVM	MLP	k-NN
TP0	0.98	0.96	0.89	0.90	0.96
TP1	0.94	0.87	0.75	0.77	0.82
TP2	0.91	0.77	0.84	0.82	0.84
TP3	0.86	0.60	0.42	0.51	0.77
TP4	0.85	0.61	0.51	0.70	0.64
LFP0	0.97	0.94	0.86	0.91	0.95
LFP1	0.90	0.88	0.84	0.86	0.80
RFP0	0.92	0.85	0.82	0.83	0.84
RFP1	0.70	0.62	0.57	0.59	0.65
RFP2	0.61	0.53	0.50	0.53	0.57
Average	0.86	0.76	0.70	0.74	0.78
Time * (ms)	45	33	138	0.25	28

* Total prediction time for the three body parts for each frame.

**Table 3 sensors-21-03346-t003:** F1 score for each posture class based on new sensor designs.

Layout	TP0	TP1	TP2	TP3	TP4	LFP0	LFP1	RFP0	RFP1	RFP2
D1	0.89 **	0.72 **	0.62 **	0.64 **	0.05 **	0.89 **	0.64 **	0.91 **	0.29 **	0.47 **
D2	0.92 **	0.69 **	0.61 **	0.67 **	0.10 **	0.90 *	0.69 **	0.91 **	0.22 **	0.50 **
D3	0.97 **	0.91 *	0.89 *	0.84 **	0.57 **	0.94 *	0.80 **	0.93 *	0.42 **	0.54 *
D4	0.98 *	0.91 *	0.85 *	0.83	0.68 **	0.94	0.81 **	0.92	0.56 **	0.53 **
D5	0.98	0.94	0.88	0.85	0.83 *	0.95	0.86 *	0.93	0.68 *	0.58 *

** *p* < 0.01, * 0.01 ≤ *p* < 0.05.

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
