# Peer review of "Exploration of Driver Posture Monitoring Using Pressure Sensors with Lower Resolution"

_sensors, 2021, doi:10.3390/s21103346_

Round 1

Reviewer 1 Report

This paper aims to explore the possibility of using pressure sensors with lower resolution for driver posture monitoring. This technology has vast implications for physical activity monitoring if properly validated. The work is interesting, however, there are a few changes that could be made to make the results more accessible and clear to readers, in details: 1. Literature review could be more focused. Maybe there lacks detailed explanations about the key contributions. The readers need more help to understand what is important, what is new, and how it relates to the state of art. 2. What are the limitations of the proposed pressure sensors based method? The author did not mention it in the paper. 3. More comparison with the literature may be provided in the experimental results section. Some examples are: - Long Liu, Zhelong Wang, Sen Qiu. Driving Behavior Tracking and Recognition Based on Multisensors Data Fusion, IEEE Sensors Journal 20(18): 10811-10823, 2020. - Gravina, R., & Li, Q. Emotion-relevant activity recognition based on smart cushion using multi-sensor fusion. Information Fusion, 48, 1–10. 2019 - Ma, C., Li, W., Gravina, R., Cao, J., Li, Q., & Fortino, G. Activity level assessment using a smart cushion for people with a sedentary lifestyle. Sensors, 17(10), 1–19. 2017 4. What are the implications of the findings? More discussion should be provided in the manuscript. The factors that influence the accuracy of driver posture monitoring should be analyzed in more detail in the discussion section. 5. Other minor concerns: -How do you deal with sensor misplacement? -Proofread the paper and improve readability.

Reviewer 2 Report

The article is well structured, and the research was properly conducted. Unfortunately, the study has very important limitations, e.g., only laboratory work and no real tests , which the authors are ware and acknowledge at the end of the manuscript. Still the article is interesting and has merit. I urge the authors to clearly refer the context and limitations of the study in the introduction section.

Reviewer 3 Report

A study about the possibility of using pressure sensors with lower resolution (cost-effective solution) for driver posture monitoring to recognize five typical torso postures, two typical left foot postures and three typical right foot postures is carried out in this paper and it is the main subject the authors deal with. Authors tested five different supervised machine learning techniques (Random Forest, Naïve Bayes, Support Vector Machine, Multilayer Perceptron and K-Nearest Neighbors) to recognize the postures of each body part and used the leave-one-out cross-validation for evaluating their performance. In this sense, the authors claim experimental results show that the Random Forest classifier outperformed the other classifiers with an average classification accuracy of 86% using the original pressure mats and 85% when only 8% of the pressure sensors were available. Authors argue this study shows the feasibility of using fewer pressure sensors for driver posture monitoring and suggests research directions for better sensor designs. In particular, the authors state that the number of sensors can be significantly reduced while keeping considerable posture recognition accuracy. However, I think that the authors should make an effort to improve the paper by taking into account the following remarks:

  • Some driver posture actual pictures should be included in Figure 3 for a better understanding of the postural class definition. Figure 3 should be improved, with more detailed information on actual images with better views of each class.

  • The characteristics (hyperparameters) of the tested classifiers should be mentioned for readers' information and analysis.

  • The authors should clarify what is for them an appropriate price for this technology, specifically, pressure-sensors-based system. How much is the "price barrier"?

  • Experiments on real driving conditions should be presented to argue “real-time driver posture monitoring” as the authors claim in the summary paragraph at the end of the conclusion section.

  • The proposed approach in this study should be adapted to different driving conditions and also be suitable for any vehicle. So, it is desirable for future work to demonstrate such driving conditions adaptability. It is also important to demonstrate the feasibility and its effectiveness with different types of vehicles (cars, busses, trucks, etc). Could there be differences in the results or not?

Round 2

Reviewer 1 Report

I value the authors efforts to answer all the previous concerns and feel that the paper overall improved.  It seems to me that most of the previous concerns were well addressed in the revised manuscript. Overall, the revised paper quality largely meets the requirements of "sensors", hence the manuscript could be accepted for publication after minor methodological errors and text editing.

Author Response

Point 1: Concerning the comments of the reviewer 3, the majority of them are well adressed and modified by the authors except the one concerning the "real-time" application, which is not convincing enough. If they plan to test the real time performance in the future work, they cannot claim the real-time performance in the conclusion paragraph.

Response 1: As requested by the editor, the authors stopped claiming the "real-time" performance in the revised manuscript because of the lack of real road test.